# Performance of Deauville Criteria in ^[18F]^FDG-PET/CT Diagnostics of Giant Cell Arteritis

**DOI:** 10.3390/diagnostics13010157

**Published:** 2023-01-03

**Authors:** Jeffrey Siefert, Jonas Kaufmann, Felix Thiele, Thula Walter-Rittel, Julian Rogasch, Robert Biesen, Gerd R. Burmester, Holger Amthauer, Udo Schneider, Christian Furth

**Affiliations:** 1Charité—Universitätsmedizin Berlin, Corporate Member of Freie Universität Berlin and Humboldt-Universität zu Berlin, Department of Nucelar Medicine, Augustenburger Platz 1, 13353 Berlin, Germany; 2Charité—Universitätsmedizin Berlin, Corporate Member of Freie Universität Berlin and Humboldt-Universität zu Berlin, Department of Radiology, Augustenburger Platz 1, 13353 Berlin, Germany; 3Berlin Institute of Health atCharité—Universitätsmedizin Berlin, Charitéplatz 1, 10117 Berlin, Germany; 4Charité—Universitätsmedizin Berlin, Corporate Member of Freie Universität Berlin and Humboldt-Universität zu Berlin, Department of Rheumatology and Clinical Immunology, Charitéplatz 1, 10117 Berlin, Germany

**Keywords:** ^[18F]^FDG-PET/CT, giant cell arteritis, deauville criteria, inter-rater agreement

## Abstract

In this retrospective study, PET/CT data from 59 patients with suspected giant cell arteritis (GCA) were reviewed using the Deauville criteria to determine an optimal cut-off between PET positivity and negativity. Seventeen standardised vascular regions were analysed per patient by three investigators blinded to clinical information. Statistical analysis included ROC curves with areas under the curve (AUC), Cohen’s and Fleiss’ kappa (κ) to calculate sensitivity, specificity, accuracy, and agreement. According to final clinician’s diagnosis and the revised 2017 ACR criteria GCA was confirmed in 29 of 59 (49.2 %) patients. With a diagnostic cut-off ≥ 4 (highest tracer uptake of a vessel wall exceeds liver uptake) for PET positivity, all investigators achieved high accuracy (range, 89.8–93.2%) and AUC (range, 0.94–0.97). Sensitivity and specificity ranged from 89.7–96.6% and 83.3–96.7%, respectively. Agreement between the three investigators suggested ‘almost perfect agreement’ (Fleiss’ κ = 0.84) A Deauville score of ≥4 as threshold for PET positivity yielded excellent results with high accuracy and almost perfect inter-rater agreement, suggesting a standardized, reproducible, and reliable score in diagnosing GCA. However, the small sample size and reference standard could lead to biases. Therefore, verification in a multicentre study with a larger patient cohort and prospective setting is needed.

## 1. Introduction

Combined Positron-Emission-Tomography/Computerized-Tomography (PET/CT) using ^[18F]^Fluorodeoxyglucose (^[18F]^FDG) is an established molecular imaging modality to visualize glucose metabolism [1]. Due to the accumulation and trapping of ^[18F]^FDG within activated inflammatory cells [2], it has been used to detect inflammation in infectious and inflammatory diseases [3]. Thus, PET-based imaging of patients suffering from giant cell arteritis (GCA) was performed as early as 1999 [4]. GCA is increasingly recognized as a systemic disease, which can involve discontinuous vascular regions of the entire body [5]. In absence of temporal involvement diagnosis can be difficult since typical symptoms of cranial involvement are commonly absent in extra cranial disease [6]. In addition, other typical symptoms such as fever, myalgia, weight loss and elevation of blood inflammatory markers are unspecific [7].

^[18F]^FDG-PET/CT has proven to be sensitive in the diagnosis of GCA [8] and may in some cases be the only modality suitable for diagnosis [9]. However, there is still a lack of standardization in interpretation of ^[18F]^FDG-PET/CT scans in patients suffering from GCA [10]. In particular, a robust, reliable, and endpoint-oriented definition of PET-positive and PET-negative has yet to be established. Thus, the data in the literature published to date are only comparable to a very limited extent; thus, limiting the widespread use of ^[18F]^FDG-PET/CT in patients suffering from GCA.

In comparison, assessment of disease activity in patients suffering from lymphoma has been successfully standardized using the Deauville criteria [11], with observations showing high degree of accuracy and inter-rater reliability [12,13]. In the Deauville system, up to 5 points are scored for the intensity of FDG enhancement. By comparing FDG enhancement in the region of interest with that of mediastinum and liver, an internal standardisation is provided.

Thus, we are proposing the Deauville criteria as a new way of diagnosing patients suffering from GCA. The applicability was evaluated by multiple readers using the criteria in patients versus a control group and assessing multiple vascular regions to evaluate robustness and reliability of the Deauville criteria. In addition, we investigated whether the applied score also allows a distinct differentiation between PET-positive versus PET-negative.

## 2. Materials and Methods

### 2.1. Study Population

This monocentric, cross-sectional study screened all patients for study inclusion who received ^[18F]^FDG-PET/CT imaging in the nuclear medicine department of Charité–Universtitätsmedizin Berlin from April 1st, 2016, to June 28th, 2020. Inclusion criteria were age >40 years, prednisolone and prednisolone equivalent doses <20 mg/d at the time of PET imaging, glucocorticoids for a duration <72 h before PET and no vascular implants). Standard of reference for patients with active disease were the proposed 2017 expansion of the ACR criteria for GCA [14]. Additionally, all patients needed to have a documented indication for initiation, escalation or change of therapy at the time of the scan, as per obtainable records. The clinical diagnosis incorporated the patients’ medical history, physical examination, clinical symptoms, laboratory tests and imaging, including PET/CT studies examined in this study. Clinicians did not receive reports based on the evaluated Deauville criteria at the time of initial diagnosis, as it was not implemented as clinical standard.

Out of 234 patients with suspected disease, the data sets of 29 patients with confirmed GCA were eligible for analysis. The patient group was matched with a control group by random selection from the patient pool with negative final diagnosis, who fulfilled the same in- and exclusion criteria, thus resulting in a patient population of 59 patients (Table 1). 

All analyses were in agreement with the Declaration of Helsinki and the ethical standards of the research ethics committee of the Charité—Universitätsmedizin Berlin (ID number: EA1/182/20; approved at 30th September 2020).

### 2.2. PET/CT Imaging

The tracer ^[18F]^FDG was used for PET imaging, and scans were performed on two PET/CT devices. Patients were advised to fast for at least 6 hrs before examination. Scans were conducted with acquisition times of 2 to 3 min per bed position and had to include at least the base of the scull to the proximal femurs.

Forty-two of the 59 patients received a scan on a digital PET device (General Electric^®^ Healthcare, Discovery MI, Chicago, Illinois, United States; silicon photomultipliers [SiPM], 3-ring detector setting, Time of flight [TOF] capability, system sensitivity of 7.3 cps/kBq). Median injected activity equalled 279.5 MBq (range, 227–397 MBq) with a median uptake time of 67.5 min (range, 58–111 min). PET raw data were reconstructed iteratively with Bayesian penalized likelihood reconstruction (GE “Q.Clear”) with a penalization factor β of 450 (matrix, 256 × 256; voxel size, 2.73 × 2.73 × 2.78 mm^3^).

In 17/59 patients, scans were performed on a non-digital PET -device with photomultiplier tubes (Philips^®^, Gemini TF16, Eindhoven, The Netherlands; lutetium-yttrium oxyorthosilicate scintillation crystals, TOF technology, system sensitivity of 6.6 cps/kBq). Median injected activity equalled 252 MBq (range, 228–300 MBq), median uptake time was 72 min (range, 58–120 min). PET raw data were reconstructed using iterative reconstruction (ordered subset expectation maximization; OSEM) with TOF analysis (BLOB-OS-TF; iterations, 3; subsets, 33; filter, ‘smooth’ [kernel width, 14.1 cm; relaxation parameter, 0.7]; matrix, 144 × 144; voxel size, 4.0 × 4.0 × 4.0 mm^3^).

With both PET scanners, non-enhanced low-dose CT data served for attenuation correction. Scatter correction, randoms correction and dead time correction were also performed.

Values for injected activity, uptake time and blood glucose separated by diagnosis can be found in Table 1.

### 2.3. Image Assessment

Imaging data were assessed independently by three readers with varying level of experience (reader1 [R1], >10 years; reader2 [R2], >3 years; reader3 [R3], <6 months). Seven-teen vascular regions (aorta with its main branches; ascending, thoracic, and abdominal aorta plus aortic arch; assessed branches included the innominate artery as well as carotid, vertebral, subclavian, axillary, iliac and femoral arteries) were scored individually per patient. When the arterial regions were not fully displayed (e.g., femoral arteries) the visible areas were scored. The highest scored lesion was decisive for the patient-based analysis. The readers were blinded to the clinical information and the diagnosis of the patients.

Readers were trained in the application of the Deauville criteria (Figure 1) by reference training data sets not included in the final analysis.

Data sets were analysed visually with the axial PET images. However, readers were asked to measure exemplary standardized uptake values (SUVmean) of the respective vessel segment using 2D-ROIs after assigning the Deauville scores. The points of measurement were chosen by the readers as per highest visual uptake in each segment.

Furthermore, each reader had to perform a standardised measurement of the SUVmean in the liver (right liver lobe) and in the right cardiac atrium (blood pool). The liver and cardiac volumes of interest (VOI) should only include areas with physiological tracer uptake; in particular, in the area of the right atrium, parts of the myocardium should not be included by the VOI. The SUV-values were measured for internal quality control, and to correct for visual phenomena. Fusion data sets should only be used for anatomical validation of the respective vessel segment. No specific pre-sets for window settings were used.

### 2.4. Statistical Analysis

Computations were primarily done using SPSS 27 (IBM Corporation, Armonk, NY, United States). Histograms, Q-Q plots, and Shapiro–Wilk-Test concluded a non-parametric distribution of data. Unless otherwise mentioned, descriptive results are presented as median, interquartile range (IQR), and range. Further calculations for receiver operating characteristics (ROC) with areas under the curve (AUC) and their 95%-confidence interval (95%-CI) were performed. Optimal cut-off values were determined based on the minimal distance *d* to the point (0, 1), defined as follows:$$\sqrt{{\left(1-Sensitivity\right)}^{2}+{\left(1-Specificity\right)}^{2}}$$

Sensitivity, specificity, positive predictive value (PPV), negative predictive value (NPV), and accuracy for cut-off points were calculated using commonly known equations, while confidence intervals were calculated with an excel-tool from “ACOMED Statistik” [Dr. Thomas Keller, Leipzig, www.acomed-statistik.de (accessed on 19 December 2021)].

To compare two readers regarding their accuracy, McNemar’s test for paired nominal data was used.

Furthermore, variability between paired readers in evaluating each vascular region as well for diagnosing patients was computed using Cohen’s Kappa (κ). For investigating overall agreement, Fleiss’ κ was calculated. Computations were done by classifying the highest Deauville-based assessment of all vascular regions as positive or negative for GCA, as defined by a chosen threshold. To classify and interpret κ -values, the benchmarks by Landis and Koch [15] were used. Confidence intervals for κ were calculated like McHugh [16] proposed, in order to compare confidence intervals to investigate significant difference between the pairs of reviewers. Statistical significance was assumed at α = 0.05.

## 3. Results

### 3.1. Patient Characteristics

In the elected time frame, 234 patients with suspected vasculitis received PET/CT imaging. 35 of whom fulfilled the revised 2017 criteria for GCA. Because of glucocorticoid treatment above the limit (n = 2), image quality (n = 2), myocarditis (n = 1) and vascular implants (n = 1) patients had to be excluded resulting in a total of 29 patients with GCA. The patient with myocarditis was excluded due to possible affection of the ascending aorta. In 25 of the 29 patients, the initial diagnosis was made during the inpatient workup. In 4 cases, PET/CT was performed due to a relapse.

11/29 patients showed purely nonspecific symptoms, such as fatigue, fever, night sweats or weight loss. Ten others presented with joint or muscle pain in addition to non-specific symptoms. Two patients showed myalgia or arthralgia. Only 4/29 patients exhibited specific symptoms of cranial GCA, such as jaw claudication or loss of visual acuity. Twelve patients exhibited unspecific cranial symptoms such as headaches or dysphagia, with two patients being diagnosed after apoplexy workup. One patient was described as oligosymptomatic. In the control cohort, the most often described symptoms were myalgia or arthralgia (20 patients), overlapping with non-specific symptoms (fatigue, fever, night sweats) in 19 patients. Final diagnoses of the control group were rheumatoid arthritis (n = 8), connective tissue diseases (n = 4), idiopathic serositis (n = 3), inflammation of unknown origin (n = 3), gout arthritis (n = 2), sarcoidosis (n = 2), gastrointestinal disease (n = 2), malignancy (n = 1), reactive arthritis (n = 1), Adult-onset Still’s disease (n = 1), polymyalgia rheumatica (n = 1) ANCA vasculitis (n = 1) and hypertensive crisis (n = 1).

In addition to PET/CT, other imaging studies performed during clinical workup were (20/59), MRI (13/59), or CT (7/59). Temporal artery biopsy was performed in 2/29 patients, with confirmation in one patient. 22/59 patients did not receive any relevant vascular imaging.

### 3.2. Diagnostic Performance

In a per patient analysis, R1 scored patients with a Deauville score of 5 (n = 8), 4 (n = 24), 3 (n = 25), 2 (n = 2), and zero patients with a maximum score of 1. R2 scored 5 (n = 22), 4 (n = 11), 3 (n = 19), 2 (n = 7), as well no patient with a score of 1. R3 differed with 5 (n = 10), 4 (n = 17), 3 (n = 5), 2 (n = 12) and 1 (n = 15).

Leading lesion-based ROC analysis revealed that a Deauville score of 4 or higher was clinically the most accurate in distinct differentiation of patients from controls (Figure 2). Highest diagnostic accuracy was achieved by R3 with 93.2%, resulting in a sensitivity and specificity of 89.7% and 96.7% (PPV, 96.3%; NPV, 90.6%), with an AUC of 0.97 (95%-CI, 0.94–1.0). Results for R1 and R2 were similar, but concluded higher sensitivities, while accuracy and specificity were not as high as in R3. Exact values, results on PPV, NPV and respective 95%-CIs can be found in Table 2. R2 and R3 differed significantly in accuracy (*p* < 0.05), while the pairs with R1 (R1/R2 and R1/R3) showed no significant differences in accurate diagnosis (*p* > 0.05).

In a case-by-case analysis, no patterns in false positive or false negative findings emerged.

### 3.3. Comparison with Cut-Off ≥3

While a cut-off of ≥4 resulted in best clinical outcome in the patient cohort, results for an alternative cut-off ≥3 were also analysed. In R3, accuracy with cut-off ≥3 remained similar with no significant difference (*p* > 0.05) at 91.5% (95%-CI, 81.3–97.2%), but sensitivity increased from 89.7% (95%-CI: 83.5–98.1%) to 96.7% (95%-CI: 82.2–99.9%) and specificity decreased from 96.7% (95%-CI: 82.8–99.9%) to 86.7% (95%-CI: 69.3–96.2%). For R1 and R2 accuracy decreased significantly (*p* < 0.05) from 91.5% (95%-CI: 81.3–97.2%) to 52.5% (95%-CI, 39.1–65.7%) and from 89.8% (95%-CI: 79.2–96.2%) to 61.0% (95%-CI: 47.4–73.5%). Sensitivity was 100% (95%-CI: 88.1–100%) for both readers while specificity decreased to 6.7% (95%-CI: 0.8–22.1%) and 23.3% (95%-CI: 9.9–42.3%).

### 3.4. Inter-Rater Agreement

When comparing the highest scoring vascular regions, agreement was insufficient. However, in dichotomizing the criteria in PET positive ( ≥4) and negative (1–3), readers achieved substantial to almost perfect agreement [15].

Most concordant ratings were achieved by the pair R1/R2 with 56/59 (94.9%) matching reads, resulting in κ = 0.90 [95%-CI, 0.76–1.0 (almost perfect agreement)]. R1/R3 diagnosed 54/59 (91.5%) patients equally with κ = 0.83 [95%-CI: 0.65–1.0 (almost perfect agreement)]. The pair R2/R3 achieved 53/59 (89.8%) equal reads, κ = 0.80 [95%-CI, 0.64–0.96 (substantial agreement)]. Because of overlap in confidence intervals, no significant difference between the pairs of readers can be assumed.

In 52 subjects (88.1%), concordant Deauville-based diagnoses were observed between all three readers, two of whom were concordant between readers but deviated from the reference diagnosis. Overall, resulting in Fleiss’ κ of 0.84 [95%-CI, 0.69–0.99 (almost perfect)].

## 4. Discussion

GCA is the most common of the vasculitides [17] and, if not detected and treated in a timely and properly manner will become a devastating disease by causing permanent damage. The diagnostic delay for GCA is prolonged if cranial symptoms are absent [18], due to the unspecific symptoms in which the disease manifests. In these cases, ^[18F]^FDG-PET/CT has shown potential as a valuable diagnostic tool [10,19]. However, the widespread use of ^[18F]^FDG-PET/CT remains limited by the lack of standardization of image interpretation and the absence of reliable and reproducible diagnostic criteria [10,20,21].

Therefore, we aimed to evaluate a standardized, PET-based approach for diagnosis of GCA—*the Deauville criteria* [11]. Choosing this score was based on its clear and concise definition of ^[18F]^FDG uptake patterns with reference to areas of physiological uptake as well as its proven high interobserver reliability for patients suffering from Hodgkin- and non-Hodgkin lymphomas [13,22,23]. Hence, leading to its adaptation for staging and response assessment in prospective studies in lymphoma patients [24,25,26] and ultimately to the use of ^[18F]^FDG-PET/CT as standard of care in clinical routine [27]. Thus, we evaluated whether this dedicated score is robustly and reliably applicable in patients with GCA. In addition, the clinical relevance of possible thresholds in to differentiate PET-negative versus PET-positive results was analysed.

We found that this standardized approach yielded excellent results in diagnostic performance with an average accuracy of 91.5%, sensitivity of 94.3% and specificity of 88.9%. In comparison, two meta-analyses from Besson et al. in 2011 [9] and Lee et al. in 2016 [28] reported inferior results to our study with sensitivities of 80% and 83.3% and similar specificities of 89% and 89.6%. However, contrary to our results, previous studies reported higher specificity than sensitivity [29,30,31,32]. In the context of everyday clinical practice, the higher sensitivity achieved in the current analysis could improve the therapeutic process and avoid under-treatment. 

Further analyses concluded an overall agreement of Fleiss’ κ of 0.84, which corresponds to an almost perfect agreement [15]. Despite the different levels of experience, no significant differences between the pairs of reviewers were detected, which we attribute to the standardized approach. Differences in measurement between the readers can largely be explained by methodical deficits of the Deauville criteria, as scores of 4 and 5 are not clearly defined, as well as variance in the leading lesions chosen by visual analysis. Compared to magnetic resonance angiography [33] with a Cohen’s κ = 0.73, agreement seems to be favourable in PET/CT, concordant with the results of Meller et al. who already demonstrated superiority of PET over MRI in early stages of inflammation in 2003 [34]. 

As early as 2015, Lensen et al. [35] reported superb agreement for extra cranial GCA in ^[18F]^FDG-PET/CT with a κ = 0.96 for 4 readers, using an uptake higher than liver uptake as visual threshold. The average sensitivity and specificity were 100% and 98%, respectively. Other tested methods were diagnosis by first impression as well as differing visual thresholds, namely, higher than the femoral artery uptake, or equal to the liver uptake. These approaches, however, led to inferior results, strengthening the argument for a visual threshold higher than the liver uptake. However, these results are weakened as the reported outcome was calculated after a consensus meeting, prior to which up to 1/3 of diagnoses were discordant [35]. The prior consensus meeting and the difference between the two examiners with least experience (two years compared to less than six months in our study) suggest that the aforementioned study is less reflective of a clinical real world setting.

Although PET/CT has been playing an important role in GCA diagnostics for many years, no real standardization and no definite cut-off between PET-positive and PET-negative could be established, and even most recent recommendations (joint recommendation of the EANM, SNMMI and PIG) remain indefinite [10]. For that reason, numerous systems have been developed, but neither semi-quantitative nor visual grading systems using either liver or blood pool activity as reference have been able to establish themselves [36]. In contrast to a proposed 4-point scale where uptake at liver level is classified as possibly PET-positive, the Deauville criteria only differentiate between uptake higher or lower than the liver uptake. For vessel wall lesions located close to the physiological reference regions (e.g., liver, and mediastinal blood pool), comparison of ‘equal to liver uptake’ may be possible. However, with larger distances between lesion and reference structures, background uptake perceptual phenomena, like the Mach effect [37], become relevant, making grading of ‘equal to liver’ almost impossible. As this phenomenon is also known from the validation phase of the Deauville criteria for the use of ^[18F]^FDG-PET in patients with lymphoma [38,39] it is known that it can be partially circumvented by giving the liver uptake a semi-quantitative value, which then corresponds to a punctual threshold—the standardized uptake value.

This is a possible explanation of the high interobserver agreement in our study, and why no consensus read was necessary. Overall, this perceptual phenomenon may explain the slight advantage in accuracy in our analysis and Lensen et al. (2015) [35] in comparison to the meta-analyses by Besson et al. (2011) and Lee et al. (2016) [9,28].

Our study deals with several limitations which must be considered. Most importantly, the small number of patients included in the study. Due to the rarity of the disease and the unspecific symptoms, only a small patient collective with confirmed GCA has emerged from the much larger collective with suspicion of GCA. Another factor in the small sample size are the in- and exclusion criteria, which were based on the conditions of usefulness for a PET examination. However, the final patient cohort and control cohort were not significantly different in CRP, further indicating the robustness and reliability of our conclusions.

Due to the retrospective design, incubation times were not standardized. Studies have shown, that prolonged uptake times lead to better target to background ratios in assessment of vascular pathologies [40,41]. Since the liver has shown to be the superior reference structure in this study, we do not expect a significant influence of the varying uptake times on the results.

In addition, selection bias cannot be excluded, as ^[18F]^FDG-PET/CT is not the first line imaging modality and is only recommended when no other imaging, such as color-coded ultrasound, leads to conclusive results [42]. Therefore, most patients in our analysis will not have shown typical symptoms of GCA, thus the 1990 ACR criteria were not appropriate for our population as they are limited to cranial involvement [43].

Due to the aforementioned reasons, and due to the resolution limits of the scanners used the cranial vessels were not evaluated. An extension of the 1990 classification criteria has been proposed [14] reflecting that the inclusion of imaging modalities has reached clinical routine years ago but may differ depending on local availability, Since the extended criteria have already been used in large clinical trials [44], we elected to choose the 2017 revised criteria as our standard of reference. However, the chosen reference standard is partially based on imaging data, including ^[18F]^FDG-PET/CT, which reduces the general transferability of the results. 

As the Deauville criteria differentiate five scores in metabolic activity, its application may not be limited to discrimination between PET-positive versus PET-negative results at the time of baseline imaging but may also have predictive value. Huang et al. (2021) [45] have demonstrated a prognostic value of a Deauville score of 5 for true disease progression at end of treatment in extra-nodal natural killer/T-cell lymphoma. Therefore, discrimination of scores 4 and 5 as proposed by Barrington et al. (2014) [46] may prove its value in differentiating patients at higher risk of vascular complications or relapse. To pursue this issue, further investigations are needed with larger sample sizes in a prospective setting.

Furthermore, a to be established validated standardized PET-reading could serve as a read-out parameter for successful therapy in GCA in clinical studies as well as in clinical routine.

## 5. Conclusions

The Deauville criteria, known from lymphoma assessment, appear to be applicable also in patients with GCA. The score is robust and reliably applicable. Diagnostic accuracy to detect patients with GCA was similarly high at different levels of experience. Vessel wall uptake with a Deauville score of ≥4 (i.e., higher than the physiological uptake of the liver) indicated disease in almost all cases. Because of this standardized evaluation approach, the application within clinical trials using a multicentre setting appears attractive. To what extent the results shown are also transferable to the situation of response assessment cannot be answered by the present data and requires additional analysis in further, preferably prospective, studies.

## Figures and Tables

**Figure 1 diagnostics-13-00157-f001:**
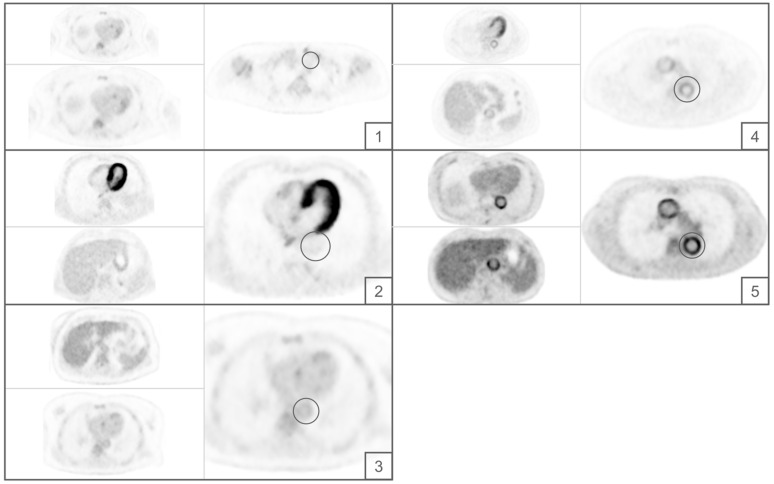
Visual representation of the Deauville criteria. Numbers 1 to 5 represent the Deauville score for each marked vessel. 1 equals no uptake. 2 equals uptake less than the mediastinal blood pool. 3 equals uptake higher than uptake in the mediastinal blood pool but lower than liver uptake. 4 equals uptake higher than liver uptake. 5 equals uptake distinctly higher than liver uptake.

**Figure 2 diagnostics-13-00157-f002:**
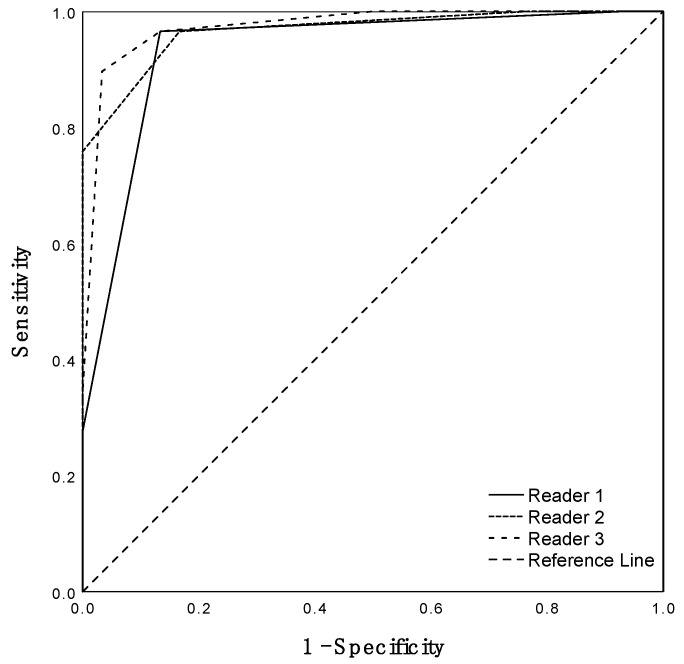
Patient-based ROC analysis per reader ROC curves in a per-patient analysis for all three readers. Calculation based on the highest Deauville score assignment for any vessel per patient.

**Table 1 diagnostics-13-00157-t001:** Description patient cohort.

Variable	Patient Cohort	Control Cohort
Total (n)	29	30
Female (n)	16/29	18/30
Male (n)	13/29	12/30
Age (years)	66 (range, 51–83)	66 (range, 49–79)
Activity (MBq)	269 (IQR, 245–302.5)	274.5 (IQR, 251–300.25)
Uptake time (min)	67 (IQR, 64–80)	71.5 (IQR, 65–84)
Blood glucose (mmol/L)	5.4 (IQR, 4.9–6)	5.6 (IQR, 5–6.3)
CRP (mg/L)	36.9 (IQR, 18.95–65.1)	38.15 (IQR, 3.8–84.9) *

Description of analysed cohort, in numbers for total patients/control patients in the examined cohort, as well as ratio female to male. Age, administered activity, uptake time between administration of ^[18F]^FDG-PET/CT imaging as well as blood glucose are given as median and respective IQR. ‘n’, count; MBq, Megabecquerel; IQR, interquartile range. * CRP not documented in n = 1 case.

**Table 2 diagnostics-13-00157-t002:** Diagnostic performance and ROC analysis.

Reader	TP	TN	FP	FN	Accuracy		Sensitivity		Specificity		PPV		NPV		AUC	
					%	95%-CI	%	95%-CI	%	95%-CI	%	95%-CI	%	95%-CI	%	95%-CI
1	28	26	4	1	91.5	81.3–97.2	96.6	82.2–99.9	86.7	69.3–96.2	87.5	71.0–96.5	96.3	81.0–99.9	0.94	0.87–1.00
2	28	25	5	1	89.8	79.2–96.2	96.6	82.2–99.9	83.3	65.3–94.4	84.9	68.1–94.9	96.2	80.4–99.9	0.97	0.93–1.01
3	26	29	1	3	93.2	83.5–98.1	89.7	72.7–97.8	96.7	82.8–99.9	96.3	81.0–99.9	90.6	75.0–98.0	0.97	0.94–1.01
Average					91.5		94.3		88.9		89.6		94.3		0.96	

TP, true positives; TN, true negatives; FP, false positives; FN, false negatives; PPV, positive predictive values; NPV, negative predictive values; AUC, area under the curve from receiver operating characteristics; 95%-CI, 95%-confidence interval.

## Data Availability

The data presented in this study are available on request from the corresponding author. The data are not publicly available due to privacy and ethical limitations.

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
