# Peer review of "Performance of Deauville Criteria in [18F]FDG-PET/CT Diagnostics of Giant Cell Arteritis"

_diagnostics, 2023, doi:10.3390/diagnostics13010157_

Round 1
Reviewer 1 Report
This article discusses the application of FDG PET-CT Deauville five-point-scaling in active giant cell arteritis. The subject is interesting and poses a diagnostic challenge for these patients: under-treatment or over-treatment are both of particular consequences. Although retrospective on a small number of patients, this study highlights some advantages of Deauville scoring. Interestingly, CRP in the comparison cohort was as high as in the patient cohort: this fact increases robustness of conclusions.
Methods:
Please define Deauville scoring more precisely, which is visual. So what are the Results about? The visual Deauville analysis or the Deauville scoring improved by using the meanSUVs on the vessel segment, liver and right atrium, as described at the end of 2.3 Image assessment? The discussion seems to confirm the use of SUV (lines 300 to 306). Please explain.
Results:
Please explain why myocarditis was a reason for exclusion a posteriori.
Define RZA, PMR and TAB.
Figure 1:
Deauville 3 training data set: why not choose the ascending aorta, which has an apparently higher FDGF uptake than the descending aorta?
References:
Cited twice: 9 and 28
Besson FL, Parienti JJ, Bienvenu B, et al. Diagnostic performance of (1)(8)F-fluorodeoxyglucose positron emission tomog-401 raphy in giant cell arteritis: a systematic review and meta-analysis. Eur J Nucl Med Mol Imaging 2011;38(9):1764-72. doi: 402 10.1007/s00259-011-1830-0
Author Response
Dear Reviewer 1,
thank you very much for your considerate feedback.
For our revisions please see the attachment.
Kind regards

Reviewer 2 Report
The method by which the physicians chosen the size and location of the ROIs in the different anatomical districts should be improved
Since the images evaluation is not completely subjective, but the choice is guided by a semiquantitative index (SUV), the authors should better justify the differences in evaluation between the readers with some examples
Author Response
Dear Reviewer 2,
thank you very much for your considerate feedback.
For our revisions please see the attachment.
Kind regards
Jonas Kaufmann
